# The Effects of Irradiation on Structure and Leaching of Pure and Doped Thin-Film Ceria SIMFUEL Models Prepared via Polymer-Templated Deposition

Alistair F. Holdsworth [1,2,*], Zizhen Feng [1], Ruth Edge [3], John P. Waters [4], Alice M. Halman [5,†], David Collison [1,*], Kathryn George [2], Louise S. Natrajan [1,*] and Melissa A. Denecke [1,3,‡]

1   Department of Chemistry, The University of Manchester, Oxford Road, Manchester, Greater Manchester, M13 9PL, UK
2   Department of Chemical Engineering, The University of Manchester, Oxford Road, Manchester, Greater Manchester, M13 9PL, UK; kathryn.george-2@manchester.ac.uk
3   Dalton Cumbrian Facility, The University of Manchester, Westlakes Science Park, Moor Row, Cumbria, CA24 3HA, UK; ruth.edge@manchester.ac.uk
4   Department of Earth Sciences, The University of Manchester, Oxford Road, Manchester, Greater Manchester, M13 9PL, UK; john.waters@manchester.ac.uk
5   School of Physical Sciences and Computing, University of Central Lancashire, Preston, Lancashire, PR1 2HE, UK; alice.halman@sellafieldsites.com
*   Correspondence: alistair.holdsworth@manchester.ac.uk (A.F.H.); david.collison@manchester.ac.uk (D.C.); louise.natrajan@manchester.ac.uk (L.S.N.)
†   Current address: Sellafield Ltd., Sellafield, Seascale, Cumbria, CA20 1PG, UK.
‡   Current address: International Atomic Energy Agency (IAEA), Department of Nuclear Sciences and Applications, Division of Physical and Chemical Sciences, Vienna International Centre, A-1400 Vienna, Austria.

**Abstract:** When studying hazardous materials such as spent nuclear fuel (SNF), the minimisation of sample volumes is essential, together with the use of chemically-similar surrogates where possible. For example, the bulk behaviour of urania ($UO_2$) can be mimicked by appropriately-engineered thin films of sufficient thickness, and inactive materials such as ceria ($CeO_2$) can be used to study the effects within radioactive systems used to fuel nuclear fission. However, thin film properties are sensitive to the preparative method, many of which require the use of highly toxic precursors and specialised apparatus (e.g., chemical vapour deposition). To address this, we present the development of a flexible, tuneable, scalable method for the preparation of thin-film $CeO_2$ SIMFUEL models with a thickness of ≈5 μm. The effects of γ irradiation (up to 100 kGy) and dopants including trivalent lanthanides ($Ln^{3+}$) and simulant ε-particles on the structure and long-term leaching of these systems under SNF storage conditions were explored, alongside the context of this within further work. It was found that the sensitivity of $CeO_2$ films to reduction upon irradiation, particularly in the presence of simulant ε-particles, resulted in increased leaching of Ce (as $Ce^{III}$), while trivalent lanthanides ($Nd^{3+}$ and $Eu^{3+}$) had a minimal effect on Ce leaching.

**Keywords:** spent nuclear fuel; SIMFUEL; thin films; irradiation; dopant effects; leaching; characterisation

## 1. Introduction

The radiological and chemical hazards associated with highly radioactive materials such as spent nuclear fuel (SNF) mean that computational methods [1] and low-activity or inactive surrogate materials are often employed to grant an understanding into the behaviour of SNF without risking exposure to or necessitating the strict procedures and equipment for handling such materials directly [2].

SNF can be analysed through remote operation in heavily-shielded hot cells including via non-destructive tests such as gamma spectroscopy [3] to determine radioisotope concentrations, and destructive ones [4] such as chemical analyses [5,6] and electron

microscopy [7] for structural analyses [8] following sufficient post-reactor cooling time. Such measurements serve to validate and improve the predictions of fuel performance codes such as ORIGEN, FISPIN [1], and derived fuel cycle simulations [8].

With currently around 300,000 tonnes of SNF accumulated globally [9] and a limited capacity to recycle or reprocess SNF, understanding its behaviour in storage has never been more important [10], especially for the development of appropriate chemical and environmental models of such systems [11]. Synthetic or simulated nuclear fuels (SIMFUELs) have been used for many years to better understand the numerous properties of urania ($UO_2$) and other fuel systems commonly utilised for the generation of power using nuclear fission [12]. These include, but are not limited to, the dissolution and leaching [13–17] and oxidation of SNF ceramics in wet and dry storage conditions [15,17] in addition to the effects of irradiation and burnup on the chemical structure [18] and physicochemical [19,20] and thermodynamic properties [21].

While the majority of SIMFUEL systems are based on $UO_2$ to match the fuel chemistry utilised in most reactor types (pressurised water reactors, boiling water reactors, advanced gas-cooled reactors, CANDU (CANada Deuterium Uranium) reactors, etc.), non-radioactive surrogate compounds can provide an alternative in situations where large amounts of active materials cannot be easily handled. Ceria ($CeO_2$) is the most common surrogate [22–24] used due to the similarity between the $MO_2$ crystal phases (both fluorite, FCC, Fm-3m) [22,25] and similar ionic radii (effective radii, 6-coordinate, $Ce^{4+}$ = 87 pm, $U^{4+}$ = 89 pm), though $ThO_2$ ($Th^{4+}$ = 94 pm) [26] has found occasional use [27] as a lower-active substituent. Doped SIMFUELs have been far more widely explored for $UO_2$ compared to $CeO_2$ and $ThO_2$ [25,28] (. Where the use of inactive materials is not suitable or possible, the ALARP principle (as low as reasonably possible/practical) [29] is often used to reduce the volume of hazardous (i.e., radioactive) substances to be handled and analysed. In the case of SIMFUELs, this can involve implementation as thin films on a suitable substrate, many of which are prepared using chemical or physical vapour deposition (CVD and PVD, also known as sputtering) techniques, which themselves require the use of rather toxic or highly reactive reagents [30,31] and specialised equipment.

While the use of thin films is beneficial in reducing the volume of material to be handled, there are often differences in chemical reactivity and physicochemical properties between film and bulk samples. Thin films of nm thickness are often used to solely study surface effects, though when appropriately engineered for sufficient thickness ($\geq \mu$m), bulk behaviour can be mimicked, depending upon the aspect of the material under investigation. Factors such as thermal conductivity are dependent on film thickness [32], while others such as dissolution, especially within the $UO_2$ system, are surface effects requiring films of sufficient thickness to adequately model the bulk system [12]. Similarly, mimicking the effects of SNF such as the formation of metallic $\varepsilon$-particles requires a shift in approach to the preparation of SIMFUELs [25]. These metallic particles form from the agglomeration of inert metal (Ru, Mo, Tc, Rh, Pd, etc.) fission products (FPs) produced during nuclear reactions, are typically sub-micron to a few microns in diameter, and can induce swelling of the fuel in addition to altering the thermal [25] and dissolution properties of SNF [33,34]. The desired analyses to be undertaken will determine the required thickness of film and thus the required preparative methods which can be employed [35], varying from high-vacuum deposition methods such as PVD [36] and atomic layer deposition (ALD) [37] to chemical approaches such as polymer-assisted deposition (PAD) [38] and layer-by-layer (LbL) approaches [39].

One of the primary motivations behind using SIMFUEL systems is to explore radiation effects on SNF ceramics, whether by ion irradiation [40] or exposure to $\gamma$ flux [13] in an effort to simulate the highly complex interactions between various highly intense radiation fluxes and ceramic materials which occur within a reactor without resorting to the aforementioned, highly radioactive, neutron irradiated SNF [41]. In other words, the study of these radiation effects combines:

- The non-ionising (thermal or fast) neutron flux which sustains the chain reaction.

- Ionising radiation:
  - The recoil effects of highly energetic FP ions following fission events;
  - $\alpha$ flux from MA (minor actinide) decay and (n, $\alpha$) reactions;
  - $\beta^-$ flux from FP (fission product) and MA $\beta^-$ decay;
  - The intense $\gamma$ emissions that accompany both of these.

The damage experienced by crystalline materials (such as SNF and SIMFUEL ceramics) upon irradiation is inversely proportional to crystallinity—with more highly ordered crystalline materials being more resistant to irradiation. Inside an operating reactor, this is combined with a constant high-temperature (1000–2000 °C) annealing effect on the fuel ceramic [41]. The level of damage inflicted upon ceramic materials by ionising radiation is dependent upon the mass of the incident particle, with the heaviest (FPs) causing the most damage but over the shortest range, and the lightest ($\gamma$ photons, being massless) cause the least damage but are highly penetrating even through dense matter [41]. The level of damage arising in ceramics thus decreases according to FP > $\alpha$ > $\beta$ > $\gamma$, resulting from the collisions of these particles or photons interacting with the electron clouds surrounding the atoms of the fuel's crystal lattice [42]. Neutrons (n) are distinct from ionising radiation in that they only interact with nuclei by scattering or capture, though this can and does cause displacement of atoms within crystal structures. Understanding this radiation-induced structural damage is imperative for the long-term safety of stored SNF in the absence of reprocessing and a closed fuel cycle [41].

In SIMFUEL systems, the damage to ceramics imparted by high-energy FPs is often mimicked via heavy ion bombardment (using e.g., Xe) in a particle accelerator, with kinetic energies approaching that observed in fission (~170 MeV) [40]. These short-ranged (order of µm) interactions from heavy ions cause lattice dislocations and ionisation along the path of travel, imparting significant heat to the ceramic; $\alpha$ particles behave in a similar manner though with a longer range as the mass is far lighter and velocity higher than other FP ions [40]. The deceleration of $\beta$ particles within ceramics (range order of cm) results in ionisation along the path of travel and the generation of bremsstrahlung X-rays; $\beta$ irradiations can be conducted using specialised electron guns [43]. The highly penetrating $\gamma$ rays which accompany $\alpha$ and $\beta$ decays damage ceramic materials via interactions including ionisations via the photoelectric effect and Compton scattering; $\gamma$ irradiations are commonly affected using specialised equipment housing $^{137}$Cs or $^{60}$Co [41].

Bearing these factors in mind and working towards the multinational development of the EURO-GANEX (European Grouped Actinide Extraction) SNF recycling flowsheet concept [44], pure- and doped ~5 µm-thick nanocrystalline $CeO_2$ thin-film SIMFUEL models were prepared using a facile, flexible, and tuneable polymer-templated deposition (PTD) method on a silica substrate. Drop-casting and low-temperature calcination were employed [45] combining elements from the works of Liu et al. [39] and Jia et al. [38]. The effects of several dopants ($Nd^{3+}$ and $Eu^{3+}$ to simulate FPs and MAs, and/or Pd nanoparticles to simulate metallic $\varepsilon$-particles) and irradiation ($^{60}$Co $\gamma$ (1.173 and 1.33 MeV) up to 100 kGy) on the structure, physicochemical properties, and leaching under interim wet fuel storage conditions of the films have been explored. The lower-crystallinity materials demonstrated here may provide a means to study model SNF systems with higher levels of damage than can be achieved with the use of bulk ceramics.

## 2. Experimental

### 2.1. Materials

Sulphuric acid ($H_2SO_4$, 98%), hydrogen peroxide ($H_2O_2$, 30% in $H_2O$), poly(diallyldimethylammonium chloride) solution (PDDA, MW > 200 000 Da, 20 wt% in $H_2O$), poly(methacrylic acid sodium salt) solution (PMAA, MW 4–6000 Da, 40 wt% in $H_2O$), ceric ammonium nitrate (CAN, $(NH_4)_2Ce(NO_3)_6$), neodymium nitrate hexahydrate ($Nd(NO_3)_3 \cdot 6H_2O$), europium nitrate hexahydrate ($Eu(NO_3)_3 \cdot 6H_2O$), tetraammine palladium(II) chloride ($[Pd(NH_3)_4]Cl_2$), and hydrazine hydrate ($N_2H_4 \cdot xH_2O$) were acquired from Sigma, VWR, or Fisher and used as received. Deionised (DI) water (>18 MΩ.cm$^{-1}$) was used for all experiments.

## 2.2. Substrate Preparation

Silica glass slides ($18 \times 18 \times 0.12$ mm) were used as the substrate. Before use, the substrates were cleaned, etched, and hydroxylated for 20 min in an acid piranha bath consisting of 3:1 by volume 98% $H_2SO_4$:30% $H_2O_2$, followed by extensive washing with DI water and drying in air at 150 °C [39]. Please note that acid piranha solution is **<span style="color:red">EXTREMELY</span>** corrosive and should be handled with utmost care. In preparation, the peroxide should be added to the sulphuric acid slowly and with sufficiently vigorous mixing to prevent potential exothermic runaway decomposition of the peroxide, which results in violent explosion of the reaction mixture.

The polymer templating solution was prepared by the rapid addition, with vigorous stirring, of a 10 g·$L^{-1}$ solution of PDDA to an equal volume 10 g·$L^{-1}$ PMAA solution, adapted from the work of Liu et al. [39]. This formed a stable, opaque white sol, which was diluted to a 4 g·$L^{-1}$ total concentration. 600 µL of this solution was deposited directly via drop-casting onto each of the etched substrates under ambient conditions using a micropipette and allowed to evaporate overnight. This volume was chosen as this was the maximum amount that could be held onto the substrate by surface tension. These slides were then heat-treated at 50 °C for 1 h in an oven before subsequent steps.

## 2.3. Synthesis of Model ε-Particles

The Pd ε-particle nanoparticle (NP) simulants used in this work were prepared via the aqueous hydrazine reduction of tetraammine palladium(II) chloride [46], a method chosen as it does not require the presence of organics or other metals to effect the reduction. $[Pd(NH_3)_4]Cl_2$ (0.25 g) was dissolved in DI water (25 mL) and added dropwise, at room temperature, with stirring, to hydrazine solution (0.37 M, 25 mL) over the course of 30 min. Once the addition was complete, the reaction mixture was refluxed for 30 min and allowed to cool to room temperature. The resultant black nanoparticles were washed extensively with DI water, dried under air at 80 °C, the yield noted (99%), and then dispersed ultrasonically in a known volume of DI water for further use. The particle size of the Pd nanoparticles was determined using a Malvern Zetasizer Nano, with a recorded size of 400 nm and narrow size distribution.

## 2.4. Preparation of $CeO_2$ Films

$CeO_2$ films were prepared via the low-temperature calcination of a CAN film, which optionally contained Nd or Eu dopants and/or Pd ε-particle simulants. For pure $CeO_2$ films, 600 µL of a 0.1 M CAN solution in DI water was deposited onto PDDA/PMAA polymer-templated slides and allowed to evaporate overnight at room temperature. Nd and Eu dopants were added to the 0.1 M CAN solution, substituting up to 5 mol% of the Ce with the respective metal nitrate. Pd ε-particles were added to the 0.1 M CAN solution from the suspension prepared previously, a known volume containing the required mass of Pd replacing up to 1 mol% of the Ce. Samples containing both Nd and Pd ε-particles were also prepared by combining these methods. These precursor solutions/suspensions were deposited and allowed to evaporate in the same way as for pure $CeO_2$. The deposited films were calcined under air in a pre-heated oven at 200 °C for 4 h, converting the CAN to $CeO_2$ and forming the final oxide films. The implications of various factors in and optimisation of the sample preparation are discussed in Section 3.1, alongside the rationale behind the approach utilised.

As the samples were prepared using a drop-casting and evaporation method, the entire metal content of the CAN/dopant solution/suspension was deposited on the silica substrate with no appreciable loss occurring during the calcination process as the metals are not volatilised in this process. This means that it is reasonable to assume that the calcined samples contained the same metal ion ratios (see Table 1) as the starting solutions used in their preparation. These were verified using EDAX elemental analysis during scanning electron microscopy (SEM) imaging.

## 2.5. Characterisation

Powder X-ray Diffraction (PXRD) patterns were collected using a Bruker D8 diffractometer equipped with a copper Kα radiation source and a 4° Si strip detector. Backscattered electron (BSE) images were collected using a Philips XL30 FEG-SEM, using an acceleration voltage of 15 kV. This apparatus was also equipped with an EDAX energy-dispersive X-ray spectrographic elemental analyser utilised for elemental mapping; it was not possible to sputter coat the SEM samples with Au or Ag as this would have interfered with further analyses. Simultaneous TGA/DSC analyses were conducted either using a Mettler Toledo STAR 1 TGA/DSC System under a 100 mL·min$^{-1}$ flow of air, with a heating rate of 10 °C·min$^{-1}$, or a Mettler-Toledo TGA1 STARe System under a 20 mL·min$^{-1}$ flow of air for isothermal studies of CAN calcination, with an initial heating rate of 20 °C·min$^{-1}$ up to an equilibrium temperature of 225, 250, or 275 °C, held for 60 min.

Differential mass analysis was performed according to previous works [47–49], with each data point calculated according to Equation (1), where $M_{diff}$ is the mass difference (%) at the temperature point, and $m_{CAN+poly}$, $m_{poly}$, and $m_{CAN}$ are the recorded masses (%) at each given temperature point for the combined polymer-CAN sample, polymer template, and CAN, respectively.

$$m_{diff} = m_{CAN+poly} - ((0.0024m_{poly} + 0.0329m_{CAN})/0.0354) \tag{1}$$

All steady state emission and excitation spectra were recorded on exposed $18 \times 18$ mm films adhered to a silica glass microscope slide using an Edinburgh Instruments FP920 Phosphorescence Lifetime Spectrometer (Edinburgh Instruments, Livingston, Scotland) equipped with a 450 W steady state xenon lamp, a 5 W microsecond pulsed xenon flashlamp (with single 300 mm focal length excitation and emission monochromators in Czerny Turner configuration), and a red sensitive photomultiplier in Peltier (air cooled) housing (Hamamatsu R928P). All spectra were reported corrected for the detector response and excitation source using the correction files supplied in the software. Lifetime data were recorded following excitation with the microsecond flashlamp using multichannel scaling. Lifetimes were obtained by tail fit on the data obtained and quality of fit judged by minimisation of reduced chi-squared and residuals squared. Only $Eu^{3+}$ emission could be observed on the samples prepared; the level of quenching was too high to measure $Nd^{3+}$.

## 2.6. Irradiation

Samples were irradiated with gamma radiation (1.173 MeV and 1.333 MeV) to an exposure of 100 kGy using a Foss Therapy 812-self-shielded $^{60}$Co irradiation source located at the University of Manchester's Dalton Cumbrian Facility. Irradiations were conducted in dry air, with an approximate dose rate of 4 k·Gyh$^{-1}$.

## 2.7. Leaching Experiments

Leaching experiments were conducted by carefully lowering a $18 \times 18$ mm film sample of each composition into separate, lidded containers bearing 50 mL of deionised water. 0.1 mL aliquots were then taken from each of these samples after 1, 2, 3, 7, 10, 14, 17, 21, and 28 days, diluted by a factor of 100 using 2% $HNO_3$, and analysed for Ce, Nd, and Eu concentrations using a Perkin Elmer NexION 450D ICP-MS (inductively-coupled plasma, mass spectrometer). Pd concentrations could not be determined quantitatively, though the counts per second (CPS) data are provided with a baseline for reference (see Figure S11). These experiments were duplicated for the irradiated samples. Repeat experiments verified the consistency of the method.

While several standardised methods for determining and analysing the leaching of nuclear materials under a wide range of environmental conditions exist [50], these were not directly compatible with the films prepared in this work. This is primarily due to the highly crystalline and/or stable nature of many nuclear materials (primarily vitrified waste glasses, but also potentially fuel ceramics) compared to the relatively low-crystalline

materials produced via the methods outlined here. With this in mind, we quantified the extent of the leaching of our samples given the rapid rate of equilibration (within 24 h, see Section 3.3 and Figures S11 and S12) observed, rather than the significantly more gradual rate of leaching values generated by the more standardised methods analysing highly crystalline and durable ceramics and glasses [51]. Some of these methods also employ dynamic or *pseudo*-dynamic systems rather than that of the batch process utilised here [51]. We refer readers to the excellent review articles by the IAEA, Ojovan and co-workers, and Thorpe and co-workers for in-depth summaries of these standard approaches [50–52].

## 3. Results and Discussion

### 3.1. Sample Preparation of CeO₂ SIMFUEL Films

As discussed in Section 1, a variety of methods can be used for the preparation of oxide thin-films such as the SIMFUEL $CeO_2$ system discussed in this work. Several approaches were assessed, most of which were unsuitable for our desired application, primarily due to undesired oxidation reactions between $Ce^{IV}$ and various organic templating agents (e.g., polyethylene imine (PEI), ethylenediamine tetracetic acid (EDTA)) during calcination, although the lack of controllability of some methods (LbL) was also a contributing factor.

A schematic summary of the film preparation is presented in Figure 1. The films were prepared by stepwise, sequential drop-casting of the components onto silica glass substrates. The substrates were cleaned and hydroxylated with acid piranha ($H_2SO_4/H_2O_2$) solution, followed by the addition of a PDDA/PMAA polymer template, the $CeO_2$ precursor, and final calcination to the desired film. This technique is facile and scalable, allowing for many samples of the same or differing compositions to be consistently prepared in parallel.

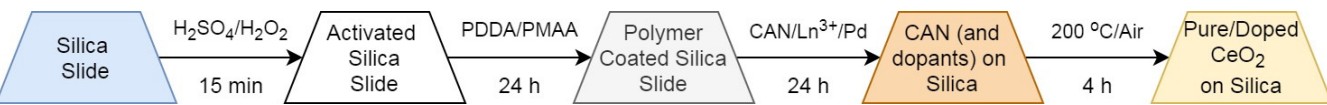

**Figure 1.** Schematic of $CeO_2$ film preparation. Processes conducted at room temperature unless otherwise specified. See Sections 2.2 and 2.4 for full details. In summary: Step 1: A 3:1 mixture of 98% $H_2SO_4$ and 30% $H_2O_2$ is used to hydroxylate the silica surface; Step 2: PDDA/PMAA polymer template solution is drop-cast and evaporated; Step 3: Film precursors are drop-cast and evaporated; Step 4: Film is calcined.

A 1:1 ratio (by mass) of PDDA and PMAA was selected as the polymer template, adapted from the work of Liu et al. [39]. This template was prepared as a sol in water at a total concentration of $4 \text{ g·L}^{-1}$, deposited as a known volume onto the silica glass substrate, and evaporated to form a film binder for the deposition of the ceria precursor. The 1:1 (by mass) PDDA/PMAA complex possesses an excess of carboxylate functionalities, which serve to coordinate metal cations in the system [39,53].

Ceric ammonium nitrate (CAN, $(NH_4)_2Ce(NO_3)_6$) is a water-soluble, stable $Ce^{IV}$ salt that has long been known to decompose upon heating under relatively mild conditions (*ca* 200 °C under air) to form $CeO_2$, proceeding via a $Ce^{III}$ intermediate [45]. The effect of temperature on the physical aspects and crystallinity of the $CeO_2$ films as measured by XRD is presented in Figures S1 and S2, highlighting the choice behind this temperature, as beyond 200 °C, aggressive oxidation of the polymer template occurs during the thermal degradation of CAN, as demonstrated in Figures S3–S5. An accompanying discussion of this rationale is presented in the Supplementary Materials.

A 0.1 M aqueous solution of CAN as the $CeO_2$ precursor was used to prepare the films in this work, deposited onto the polymer-templated silica slides and allowed to evaporate, forming an even, consistent surface. At this concentration and accounting for the volume of CAN solution applied to each sample and the density of $CeO_2$, the thickness of the produced films was around the target value of 5 µm, assuming a bulk $CeO_2$ density of $7.215 \text{ g·cm}^{-3}$.

The polymer binder is essential for a system such as that explored here due to the tendency of CAN crystallites to contract upon calcination: the cell volumes of anhydrous CAN, the $Ce^{III}$ intermediate (suspected to be $(NH_4)_3Ce^{III}_2(NO_3)_9$), and $CeO_2$ are 731.26 (731.26 $Å^3$/Ce atom), 2679.82 (553.96 $Å^3$/Ce atom), [54] and 158.78 $Å^3$ (39.7 $Å^3$/Ce atom), respectively [45,55]. CAN is monoclinic [55] while the $Ce^{III}$ intermediate and $CeO_2$ are cubic [45,56].

The thermal degradation of CAN to $CeO_2$ is essentially complete by 200 °C, though increased residence times at this temperature result in more highly crystalline samples. The requirement to achieve $CeO_2$ films of sufficient crystallinity while retaining the chemical properties of the PDDA/PMAA template required optimisation of the calcination time at 200 °C, as illustrated by the XRD patterns in Figure 2. Crystallite sizes were obtained from the FWHM (full-width at half maximum height) of the $CeO_2$ 111 peak using the Scherrer equation (cubic crystal system) XRD patterns presented in Figure 2, corresponding to a crystallite size of around 9 nm. Therefore, we selected a calcination duration of 4 h for the purposes of this work. Optical images of these samples and the respective polymer templates are shown in Figure S6. The various peaks observed in the 0.5 h and 1 h diffraction patterns presented in Figure 2 correspond to CAN.

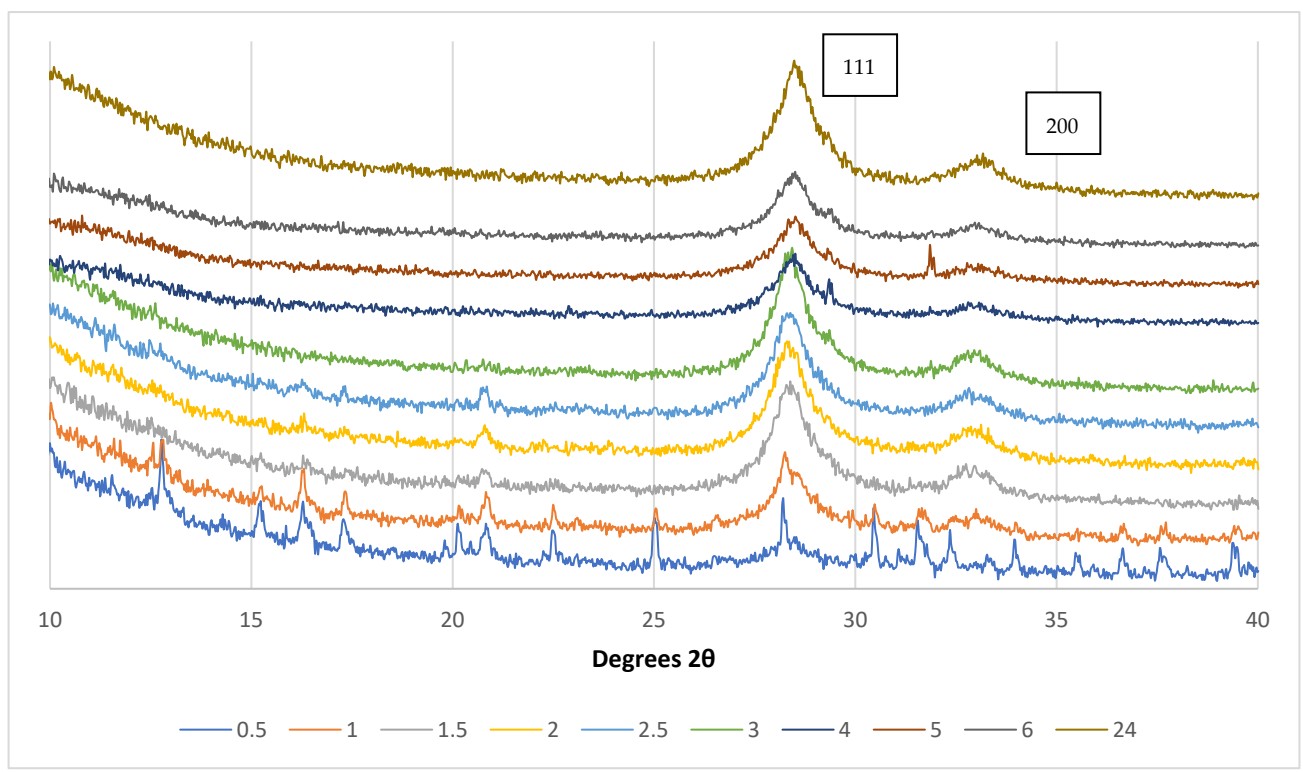

**Figure 2.** Variation in $CeO_2$ film peak formed with calcination time (in h, see figure legend) at 200 °C. Shortest times at the bottom of the graph. No post-processing was applied to these data beyond cropping.

As illustrated by the isothermal TGA analyses of pure CAN at 225, 250, and 275 °C in Figure S7, lower calcination temperatures resulted in more controlled degradation of the compound while still yielding the intended final product, $CeO_2$. Notably, the work of Audebrand et al. used heating rates of 5 °C·h$^{-1}$ in the primary degradation region of CAN [45]. However, a distinction between the thermal degradation of "fresh" crystalline CAN and the material deposited from solution (as per our film samples) will exist, and these observations can be used to support the primary data presented. A thorough study of these effects, however, was beyond the scope of this work.

The various ionic dopants representing FPs and MAs ($Nd^{3+}$ and $Eu^{3+}$) and Pd ε-particle simulants were incorporated into the films during the drop-casting of the CAN solution, substituting a proportion of the $Ce^{4+}$ ions as soluble nitrates (Nd or Eu) or suspended nanoparticles (Pd), respectively. Using this methodology, two parallel sample series were prepared, as summarised in Table 1. The Nd and Pd dopants were intended to be representative of most SNF systems, while those containing Eu could be used as surrogates for fast reactor fuels containing minor actinides.

**Table 1.** Sample compositions prepared. Percentages are expressed as mol% of total present.

| Sample | Ce (%) | Nd (%) | Pd (%) | Sample | Ce (%) | Eu (%) |
|---|---|---|---|---|---|---|
| Control | 100 | - | - | Control | 100 | - |
| Nd | 98 | 2 | - | 1% | 99 | 1 |
| Pd | 99 | - | 1 | 2% | 98 | 2 |
| Nd-Pd | 97 | 2 | 1 | 5% | 95 | 5 |

*3.2. Physicochemical Studies of CeO$_2$ SIMFUEL Films—Variations with Dopants and Irradiation*

3.2.1. Effects of Dopants and Irradiation on Film Surface Morphology

The surface morphology and distribution of metals in our film samples were determined simultaneously using SEM and EDAX mapping, respectively. As prepared, the surface morphology of our films was smooth on the micrometre level with homogeneous distribution of the metals (Ce as an example), as presented in Figure 3.

For all sample variations prepared, the distribution of EDAX-detectable elements (Ce, Nd, Eu, Pd) was uniform on a μm scale. Agglomerations of Pd NPs on or near the surface of the films could be briefly observed in SEM, but any attempt to zoom in or focus on these resulted in the rapid electrostatic charging of these particles, ejecting them from the film with some force and leaving a visible crater. Images of the full matrix of films prepared and the associated elemental maps for the Nd- and Pd-doped samples are presented in Figure S8. The Eu-doped samples were similarly homogenous.

All variations of our film samples produced were irradiated to 100 kGy using $^{60}$Co-generated γ radiation. This produced several structural and chemical changes within the samples, where the dopant ions or additive particles played a crucial role in the effects observed. The most apparent visual observation in all samples was the formation of cracks in the film surfaces, as illustrated in Figure 4. The elemental distribution within the films was unaffected by irradiation.

3.2.2. Effects of Dopants and Irradiation on Film Crystal Structures

The effects of the various dopants on the XRD patterns of our CeO$_2$ SIMFUEL films are presented in Figures S9 and S10. The pure CeO$_2$ film adopted the expected fluorite structure with a crystallite size of ~9 nm, which remained unchanged following irradiation with 100 kGy $^{60}$Co γ. For several of the doped samples, the characteristic CeO$_2$ (111) peak shifted to a slightly lower angle and indicates a mixture of crystallite grain sizes. Upon irradiation, a consistent crystallite size mirroring that of the virgin CeO$_2$ sample was observed.

The Nd, Pd, and Nd+Pd-doped samples all showed the same characteristic CeO$_2$ fluorite peaks in the XRD patterns; additional peaks were detected for the Pd-doped samples, which displayed the expected elemental Pd responses. The small peak observed at 31.7° 2θ could be attributed to nitratine (NaNO$_3$), which formed during the calcination process by a side-reaction between the film components and Na$^+$ present in the polymer binder. This could be mitigated in future work by using the commercially-available ammonium salt of polymethacrylic acid, rather than the sodium salt utilised here. These nitratine peaks become more intense in the irradiated films and appear alongside a complex series of peaks formed in the Pd-doped CeO$_2$ film upon irradiation. These could not be assigned to any known compound, but suggest that significant additional crystal phases are being formed

alongside the bulk $CeO_2$ of the films present, and these were also apparent in the 1% and 5% $Eu^{3+}$-doped films following irradiation.

| CeO₂ Film (Wide Zoom, ca. 5 mm across) | EDAX Ce elemental MAP |
| --- | --- |

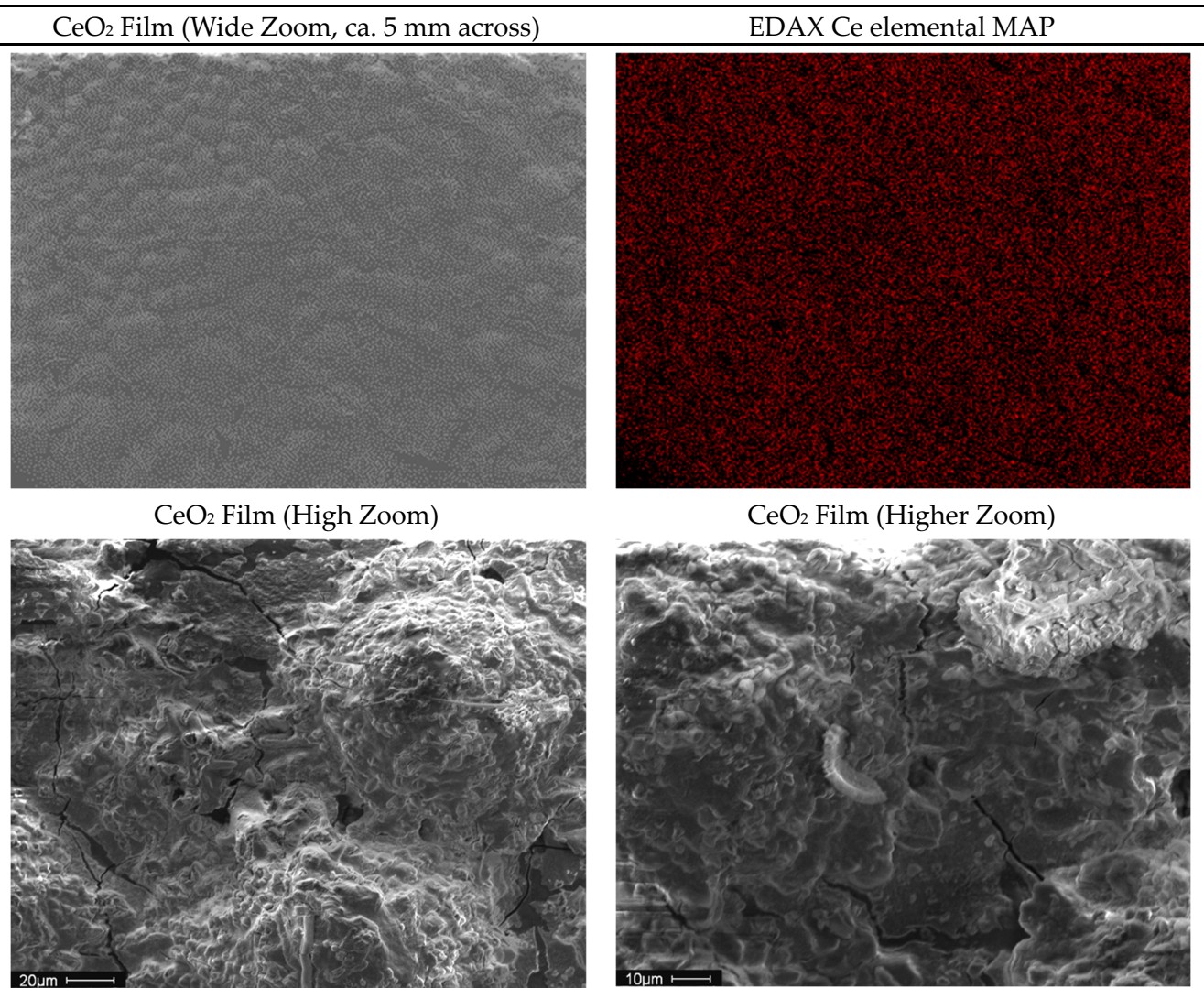

| CeO₂ Film (High Zoom) | CeO₂ Film (Higher Zoom) |
| --- | --- |

**Figure 3.** SEM images of the pure $CeO_2$ film and elemental map of Ce. The wide zoom images represent a significant fraction of the sample.

The thermal degradation of $Eu^{3+}$ and $Nd^{3+}$ nitrate salts resulted in the uniform distribution of the dopants throughout the films, either doped into the $CeO_2$ structure as $Ce_{1-x}Ln_xO_{2-x}$ where $x \leq 0.05$, as per the yttria-doping of zirconia [57], or forming separate $M_2O_3$ crystallites [58]. The thermal degradation of $Eu(NO_3)_3 \cdot 6H_2O$ releases $HNO_3$ from 25 °C, forming intermediate oxynitrates before a primary loss of $NO_2$ to the oxide [58], with neodymium nitrate behaving similarly [59]. The successful incorporation of Pd can be detected by XRD, as, despite the low concentration, the (111) peak at 39° 2θ can be detected.

Trivalent dopants (including $Nd^{3+}$ and $Eu^{3+}$) within a $CeO_2$ matrix produce oxygen vacancies that can be used to alter the properties of the material [60–62]. XRD indicated that the virgin films had doped completely, as at the dopant levels tested, both $Nd^{3+}$ and $Eu^{3+}$ were completely miscible in the $CeO_2$ matrix.

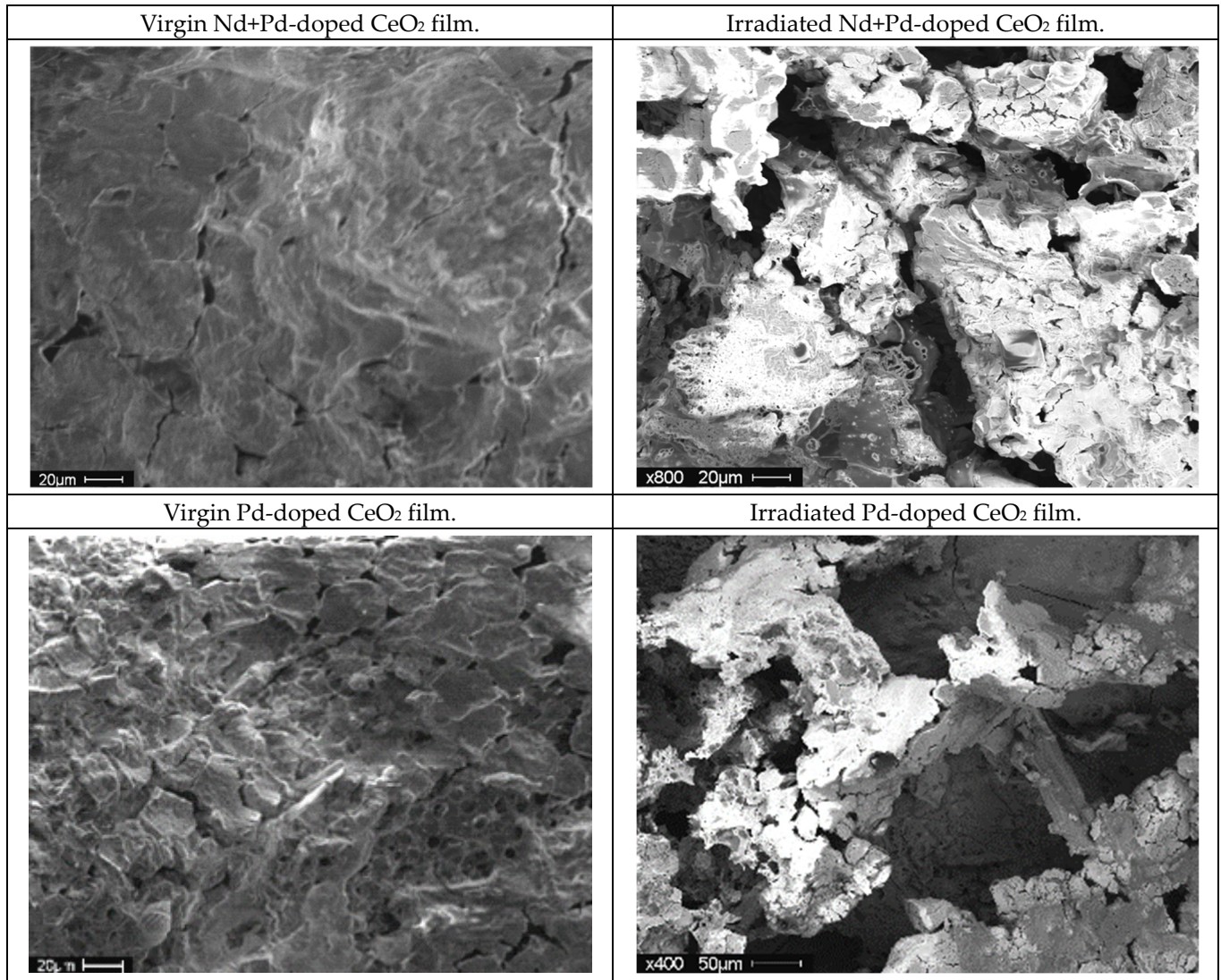

| Virgin Nd+Pd-doped CeO₂ film. | Irradiated Nd+Pd-doped CeO₂ film. |
| Virgin Pd-doped CeO₂ film. | Irradiated Pd-doped CeO₂ film. |

**Figure 4.** Effect of irradiation on the surface morphology of several doped Ce films. Note the formation of surface cracks.

### 3.2.3. Effect of Dopants and Irradiation on Chemical Environments

Steady state and time-resolved emission spectroscopy (TRES) of the $Eu^{3+}$-doped $CeO_2$ films before and after 100 kGy $\gamma$ irradiation demonstrated a negligible effect on the coordination environment or lifetime of emission decay of $Eu^{3+}$ within the samples (see Figure 5 for example), though a similar reduction in $Eu^{3+}$ emission intensity reported by Vujčić et al. was observed [63], possibly arising from radicals produced by irradiation quenching the emission and possibly forming $Eu^{2+}$. The results presented here were collected using time gated detection to eliminate the background interference caused by the polymer template. The form of the spectrum maintained by the $Eu^{3+}$ emission and the identical lifetimes (accounting for intensity, Figures 5 and S13) indicate that there was no change in the local $Eu^{3+}$ coordination environment upon irradiation and that the majority of the Eu remained as $Eu^{3+}$. The $Eu^{3+}$ emission decay parameters were determined to be biexponential (as common with solid state samples and films due to different local microstructural environments), with $t_1 = 66.24$ μs and $t_2 = 161.48$ μs for the unirradiated (5% Eu) film and $t_1 = 63.91$ μs and $t_2 = 171.41$ μs for the 100 kGy irradiated film, respectively.

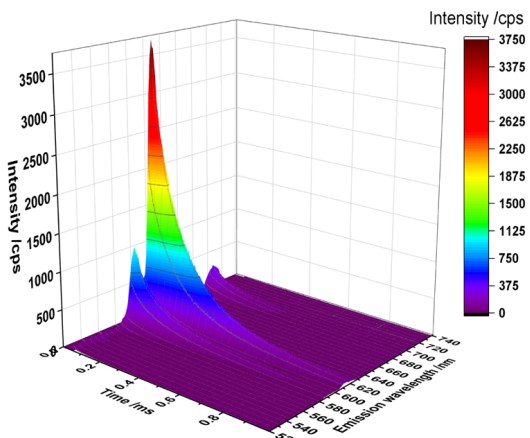 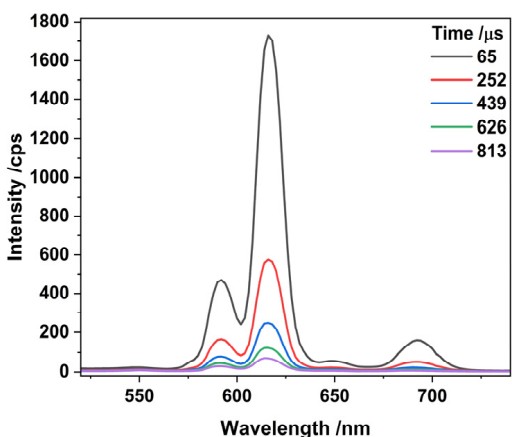

**Figure 5.** (**Left**) Time resolved emission spectrum (TRES) of the 5% Eu-doped CeO$_2$ film as prepared and prior to irradiation following 397 nm excitation with a microsecond pulsed xenon flashlamp using a 0.2 ms delay time and a 455 nm longpass filter. The 3D spectrum was constructed of lifetime decays recorded every 2 nm across the emission region of Eu$^{3+}$ (500–750 nm) and showed the typical intra configurational f-f electronic transitions of Eu$^{3+}$ ions arising from the $^5D_0 \rightarrow {}^7F_J$ (J = 1, 2, 3, and 4) transitions. (**Right**) Spectrally sliced emission spectrum from the TRES at given time points of the kinetic traces showing the decrease in the *pseudo*-steady state spectra emission intensity with time.

### 3.3. Effects of Dopants and/or Irradiation on the Leaching of CeO$_2$ SIMFUEL Films

The leaching of our film samples was assessed by exposing each variation to deionised water at room temperature for a period of 3 weeks and monitoring the evolution of metal concentrations in the liquid phase over time using ICP-MS after 1, 2, 3, 7, 10, 14, 18, and 21 days, in order to represent the conditions found in a large number of spent fuel storage ponds worldwide [64]. We believe that the observations presented below are strongly influenced by the effects of the Ce$^{III}$/Ce$^{IV}$ redox couple; Ce readily converts between these two oxidation states in aqueous environments with variations in pH and oxidation potential, with the trivalent Ce$^{III}$ state being of greater solubility [65] and higher pH values lowering the solubility of Ce$^{IV}$ more rapidly than Ce$^{III}$ [66]. The small crystallites (*ca* 9 nm) present in our film samples likely accelerate both the rate and magnitude of leaching into aqueous environments. The increased solubility of the Ce$^{III}$ oxidation state relative to Ce$^{IV}$ likely mirrors that of the U$^{IV}$/U$^{VI}$ system.

The observed concentrations are presented in Figures S11 and S12, separated by sample matrix, the metal under analysis, and the distinction between virgin and γ-irradiated examples. Complete leaching of all of each metal in a film equates to the following solution concentrations: 100% Ce: 168.13 ppm; 2% Nd: 3.46 ppm; 1% Eu: 1.82 ppm; 2% Eu: 3.65 ppm; 5% Eu: 9.12 ppm. The equilibrium metal concentrations are summarised in Table 2.

The leaching of the Nd and/or Pd-doped CeO$_2$ films did not show a simple function with respect to the dopants present within the film and the effects of irradiation. With respect to the pure CeO$_2$ films, the level of leaching was low, with ~5% of the Ce measured in the aqueous phase after 21 days increasing to 31% following 100 kGy γ irradiation. This increase in Ce leaching could arise from the radiolytically-induced reduction of the Ce$^{IV}$ centres in the films to Ce$^{III}$, radiolytic degradation of the PDDA/PMAA polymer template, or a combination of these two factors. In both virgin and irradiated systems, however, equilibrium was reached within 48 h of exposure.

**Table 2.** Summary of recorded average equilibrium metal concentrations from the pure and doped $CeO_2$ SIMFUEL thin-film leachants after 48 h, expressed as a percentage of the total metal present in the initial film that was leached.

| Sample | Ce (%) | Nd (%) | Pd (CPS) [a] | Sample | Ce (%) | Eu (%) |
|--------|--------|--------|--------------|--------|--------|--------|
| $CeO_2$ | 5 | - | - | - | - | - |
| [Irrad] | 31 | - | - | - | - | - |
| Nd | 5 | 33 | - | 1% Eu | 55 | 100 |
| [Irrad] | 9 | 33 | - | [Irrad] | 34 | 60 |
| Pd | 81 | - | 5000 | 2% Eu | 45 | 63 |
| [Irrad] | 53 | - | 4000 | [Irrad] | 23 | 36 |
| Nd+Pd | 40 | 62 | 4000 | 5% Eu | 39 | 68 |
| [Irrad] | 40 | 62 | 3600 | [Irrad] | 34 | 5 |

[a] Pd baseline was ~1000 CPS, see Section 2.7 for explanation.

The 2% Nd-doped $CeO_2$ film showed no change in the leaching of Ce upon irradiation, with an equilibrium of 9% Ce measured in the aqueous phase after 21 days. This would suggest that the presence of $Nd^{3+}$ ions serves to stabilise the $CeO_2$ structure against leaching in this system [57]. In contrast, the $CeO_2$ film doped with 1% $\varepsilon$-particle simulants showed significantly increased leaching of Ce (81% of total) after 21 days, though this was reduced (to 53% of total) upon irradiation. This phenomenon may arise from the catalytic reduction of Ce by the Pd $\varepsilon$-particles simulants present on our films, as per Equation (2). Pd $\varepsilon$-particle simulants have been reported to display catalytic behaviour in both oxidation and reduction in $UO_2$ systems [33]. In this case, we may be observing the catalytic reduction of Ce, which would explain the increased leaching where Pd is present. The measured pH for all samples was close to neutral (6.5–7.5).

$$4Ce^{4+}_{(aq)} + 2H_2O_{(l)} \rightarrow 4Ce^{3+}_{(aq)} + O_{2(g)} + 4H^+_{(aq)} \tag{2}$$

We observed a similar effect in the liquid phase—a 99 mM solution of CAN containing 1 mol% of our Pd NPs visibly lost its clear orange-yellow colouration over the course of several weeks without the formation of a precipitate, while a comparable 100 mM CAN solution maintained its original hue under the same conditions. When both $Nd^{3+}$ and $Pd^0$ dopants were combined (at 2 mol% and 1 mol%, respectively), the leaching of Ce from these films was ~40% irrespective of irradiation, suggesting a competing effect between the stabilisation provided by $Nd^{3+}$ doping and the acceleration of leaching increased by $Pd^0$.

We observed no changes to the leaching behaviour of $Nd^{3+}$ from the $CeO_2$ films upon irradiation, though the detected concentration when $Pd^0$ was also present was double that of the $CeO_2$ film doped with Nd only, with 62% and 33% leached in these scenarios, respectively. Our observations of the Nd only and Nd/Pd doubly-doped $CeO_2$ films would suggest that an equilibrium $[Nd^{3+}]$ of ~1.5 mol% in the $CeO_2$ matrix helps to stabilise the structure from further leaching, despite the generally higher solubility of $Ln^{3+}$ ions relative to $Ce^{4+}$ [65,66].

We were able to detect the presence of suspended $Pd^0$ particles during the leaching studies using ICP-MS, though the concentrations observed were below the detection limit, and hence the results presented are not quantitative as we relied on the counts per second observed, rather than quantitative measurements than could be achieved with ionic species; a baseline without $Pd^0$ is provided for comparison (see Figure S11). The results presented demonstrate that some $Pd^0$ particles are suspended during the film leaching process, though equilibrium takes longer to establish than for the ionic species. Modest variation between the virgin and irradiated samples and the Pd-only and Nd/Pd-doubly doped sample were detected, but other experimental techniques beyond the scope of this work would be required to quantify these. While in this work we utilised chemically-inert Pd nanoparticles to represent $\varepsilon$-particles as these can resist the oxidising conditions of the $CeO_2$ matrix, in $UO_2$ SNF systems, a reducing environment is encountered. The $\varepsilon$-particles that occur in $UO_2$ systems are formed from the metallic elements Mo through Ag in the Periodic

Table, while also containing some Se and Te. During nitric acid leaching, the amount of each element leached decreases from Mo to Pd, reflecting the changing chemistry over the period. Thus, the behaviour observed in real SNF systems would differ from the observations presented here [67].

In contrast to the Nd-doped samples presented above, Eu notably increased the leaching of Ce in the virgin samples. This is likely a combination of effects and cannot be readily explained by the minor difference in ionic radii (6 − coordinate $Nd^{3+}$ = 98.3 pm, $Eu^{3+}$ = 94.7 pm) [68] between the two metals or the chemistry, as the +3 oxidation state is predominant for both, although both elements do have an accessible divalent state ($Eu^{III}/Eu^{II}$ = −0.35 V, $Nd^{III}/Nd^{II}$ = −2.7 V), though both of these are unlikely to occur in the presence of $Ce^{IV}$ ($Ce^{IV}/Ce^{III}$ = +1.61 V); Nd furthermore can be oxidised (under very forcing conditions) to a tetravalent state.

The amount of Ce leached after 3 weeks of equilibration increased from 5% to 55% at 1 mol% Eu doping, with increasing Eu concentrations in the film marginally lowering the extent of Ce leaching to 45% and 39% at the 2% and 5% Eu doping levels, respectively. Following irradiation, the leaching of Ce from the Eu-doped samples was more consistent with the pure $CeO_2$ film, with 34%, 23%, and 34% Ce leached at the 1 mol%, 2 mol%, and 5 mol% Eu dopant levels, respectively. As per irradiation of the Pd samples, disruption of the Ce reduction pathway could explain the lowered Ce leaching in the irradiated samples.

The behaviour of Eu leaching was, like that of Nd, more consistent than Ce before and after irradiation, especially at the higher $Eu^{3+}$ doping levels. All of the Eu was leached from the 1% sample for the virgin sample and 60% upon irradiation, 63% and 36% for the 2% sample before and after irradiation, and 68% for both the virgin and irradiated 5% samples, respectively. As per the Nd-doped samples, the levels of Eu leaching are likely a result of the increased solubility of $Ln^{3+}$ ions relative to $Ce^{4+}$ [65,66]. Any redox effects due to the variable speciation of Eu in either of the films could not be determined as the dopant level (≤5%), and solution concentration (<9.12 ppm) precluded the use of any spectroscopic techniques capable of this kind of determination due to the limits of detection involved.

The increased leaching observed in several of the irradiated film systems likely arises from physical factors—the formation of cracks upon irradiation increase the available surface area for reaction and thus the overall extent of dissolution in the absence of more complex chemical factors impacting the leaching behaviour. In the case of the systems tested here, the increased solubility of Ce due to reduction from the $Ce^{VI}$ to $Ce^{III}$ oxidation state could be beneficial in the $UO_2$ system, where oxidation leads to the leaching of U into the aqueous phase as the $UO_2^{2+}$ cation via oxidation from the $U^{IV}$ to $U^{VI}$ states [15,17,33,67]. As the samples prepared here were irradiated "dry", the interactions between the surfaces of the film ceramic and the contained nanoparticles could not be quantified and would require differing approaches to sample preparation, irradiation, and analysis [69,70]. Such variations are beyond the scope of the work presented here.

## 4. Conclusions

The systems explored and the results presented here represent a development of SIMFUEL studies modelling aspects of events such as cladding breaches in SNF storage, though many aspects of such real-life occurrences cannot be readily modelled in a low-active laboratory setting. For example, SNF produces a constant, high-intensity radiation field from the combined decays of actinides and FPs, resulting in the radiolysis of water and the generation of a combination of oxidising and reducing species such as $H_2O_2$, $HO^{\bullet}$, and $H_2$. In reality, it is challenging to inflict the extent of damage received by nuclear fuels during reactor irradiation, despite the fact that fuel ceramics are otherwise quite resistant to the irradiation conditions used here.

As previously stated, materials of lower crystallinity such as the SIMFUEL films prepared and studied here are more susceptible to radiation damage than bulk, highly crystalline ceramics, and as such may serve as suitable mimics for highly-irradiated and highly–crystalline solids. We do acknowledge that there are limitations to the preparative

technique employed in this work such as the ~200 °C limit on calcination temperature due to the onset of degradation and the oxidising nature of the CAN to $CeO_2$ calcination process, though flexibility has been demonstrated with the ability to readily entrain a variety of dopants within the $CeO_2$ film structure. While other related film deposition methods were tested in this work, the PTD method presented here proved to be the most adaptable and provided the highest quality films relative to the PAD, LbL, and oxalate calcination approaches, for example.

Future work will explore other PTD-prepared SIMFUEL systems ($UO_2$) and chemistries (simulating virgin and irradiated sodium-cooled fast reactor MOX fuel) [71] including studies of leaching in nitric acid systems over shorter timescales than those presented here and differing irradiation conditions (e.g., $\alpha$ vs. $\gamma$ irradiation). There is significant scope for broader studies, for example, the leaching under radiolytic conditions [72], the presence of neutron absorbers (boric acid), corrosion inhibitors, and species found in groundwater and cement pore water (as per a deep geological disposal repository) [64], and the leaching under acidic conditions to more closely replicate those of SNF reprocessing, though this will be conducted at lower acidities than the 8 M $HNO_3$ used to dissolve SNF ceramics in reprocessing environments. The films prepared here dissolve almost instantaneously under such conditions.

In summary, this publication has demonstrated a reproducible polymer-templated deposition method for the preparation of pure and doped $CeO_2$ film spent nuclear fuel models of 5 µm-thickness, and explored the effects of 100 kGy $\gamma$ irradiation on the structural and leaching properties of the films in simulated aqueous spent fuel storage conditions. While structurally similar to $UO_2$, the differing chemistry of $CeO_2$ presents a number of challenges in the context of use as a spent fuel model. Despite a similar ability to accommodate dopant ions, the strongly oxidising nature of $Ce^{IV}$ compared to the reducing nature of $U^{IV}$ means that contrasting behaviour is observed, the most apparent factor being the sensitivity of the $CeO_2$ films to reduction upon irradiation, particularly in the presence of Pd $\varepsilon$-particle simulants, resulting in the increased leaching of Ce. In a $UO_2$ system, this effect may produce the opposite behaviour, helping to maintain U spent fuel in the +4 oxidation state, where solubility is lower [33,72]. There remains significant scope for further study of this system to address several gaps in knowledge surrounding the interactions between the various components of the system.

**Supplementary Materials:** The following supporting information can be downloaded at: https://www.mdpi.com/article/10.3390/jne5020011/s1. Figure S1: Effect of calcination temperature on the deposited template and $CeO_2$ films calcined under air for 30 min. Figure S2: XRD diffraction patterns of films calcined at different temperatures for 30 min. Figure S3a,b: Mass loss curves (left), and rates of mass loss (right) for the PDDA/PMAA polymer template, a combined sample, and CAN, as deposited. Figure S4: Differential mass graph for the recorded TGA results. Figure S5: Heat flow of the TGA data presented in Figure S3. Figure S6: Photograph of slides after heating with varying time at 200 °C. Figure S7: Variation in mass loss over time for pure CAN crystals held isothermally at a fixed temperature. Figure S8: Elemental maps of pure $CeO_2$, Nd, Pd, and Nd+Pd-doped films. Figure S9: XRD patterns of virgin (a—$CeO_2$ and b—Nd, Pd, and Nd+Pd-doped) and irradiated (c) $CeO_2$ films. Figure S10: XRD Patterns of virgin (a) and irradiated (b) pure and Eu-doped $CeO_2$ films. Figure S11: Ce (a and b), Nd (c and d), and Pd (e and f) leaching over time in the Ce-Nd-Pd sample matrix. Virgin (a, c, e) and irradiated (Ir, b, d, f). Figure S12: Ce (a and b), and Eu (c and d), leaching over time in the Ce-Eu sample matrix. Virgin (a, c) and irradiated (b, d). Figure S13: TRLFS spectra of the 5% Eu-doped virgin and irradiated $CeO_2$ films (a) with Eu lifetime (b). These figures are accompanied by relevant discussion where pertinent. References [73–75] are cited in the supplementary materials.

**Author Contributions:** Conceptualisation, A.F.H. and M.A.D.; Methodology, A.F.H. and M.A.D.; Formal analysis, A.F.H., L.S.N. and J.P.W.; Investigation, A.F.H., Z.F., J.P.W., L.S.N., R.E. and A.M.H.; Resources, R.E., M.A.D. and L.S.N.; Data curation, A.F.H., J.P.W., A.M.H. and Z.F.; Writing—original draft preparation, A.F.H.; Writing—review and editing, A.F.H., K.G., D.C. and A.M.H.; Visualization, A.F.H. and L.S.N.; Supervision, A.F.H., L.S.N., D.C. and M.A.D.; Project administration, L.S.N. and

M.A.D.; Funding acquisition, M.A.D. All authors have read and agreed to the published version of the manuscript.

**Funding:** This project was funded by the EU Horizons 2020 Program (GENIORS (Gen IV Integrated Oxide Reprocessing Strategy), Domain 1, WP 4, grant # 755171). S. Dobson was supported by the Nuffield Foundation summer placement scheme. We further acknowledge the support of the University of Manchester's Dalton Cumbrian Facility (DCF), a partner in the National Nuclear User Facility, the EPSRC UK National Ion Beam Centre, and the Henry Royce Institute.

**Data Availability Statement:** The data presented in this study are available on request from the corresponding authors.

**Acknowledgments:** We wish to thank the researchers at OMIC (Organic Materials Innovation Centre), M. Jennings, A. Davies, J. Yarwood, J. Moore, R. Mao, and H. Eccles for their valuable technical assistance and provision of analytical equipment and services; S. Dobson for performing some initial reactions and explorations into the sample preparation methodology; and A. Woodward for help with the luminescence data.

**Conflicts of Interest:** Author Alice M. Halman was employed by the company Sellafield Ltd. The remaining authors declare that the research was conducted in the absence of any commercial or financial relationships that could be construed as a potential conflict of interest.

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
