# Peer review of "The Effects of Irradiation on Structure and Leaching of Pure and Doped Thin-Film Ceria SIMFUEL Models Prepared via Polymer-Templated Deposition"

_jne, doi:10.3390/jne5020011_

Round 1

Reviewer 1 Report

Comments and Suggestions for Authors

The article on "The Effects of Irradiation on Structure and Leaching of Pure and Doped Thin-Film Ceria SIMFUEL Models Prepared via Polymer-Templated Deposition" is of very good quality and of great interest for nuclear research. There are many results which are all well explained and introduced. For these reasons I recommend the publication of this work with minor corrections.

- In the "preparation" section it would be preferable to specify if the final composition of the samples was determined and how.

-ligne 285 : g.cm-3 ( a point is missing)

- The SEM images in Figures 3 and 4 are much too dark, we see nothing. They should be reworked in order to clarify them.

Concerning the XRD, have you made any refinements to the diffractograms? If this is the case, the lattice parameters should be provided in the form of a table as additional information. You could also compare these values to the theoretical value of well crystallized CeO2. You could thus have possible information on the presence of Ce(III) or the incorporation of Nd and Eu in the fluorine structure.

- Concerning the leaching part, the increase in the dissolution rate of the irradiated compound (31%) is in fact perhaps due to the presence of Ce (III) but it is also probably due to the presence of cracks and therefore to an increase in the reactive surface of the sample.

Generally speaking, when we look at Figures S11 and S12, the release of cations into solution takes place during the first day. This pulse (already seen in the literature) is generally due to the presence of a poorly crystallized phase (therefore very soluble), which is very likely for samples calcined at 200 °C. I think it would be better to add a few words on this effect in the text. Indeed the difference in behavior between doped Nd and Eu is very surprising, do they have the same crystallization state initially?

Author Response

We thank the reviewer for taking the time to review our work and for their helpful suggestions.

- In the "preparation" section it would be preferable to specify if the final composition of the samples was determined and how.

Clarifying statement added in Section 2.4.

-ligne 285 : g.cm-3 ( a point is missing)

Format changed throughout the manuscript and SI ass requested.

- The SEM images in Figures 3 and 4 are much too dark, we see nothing. They should be reworked in order to clarify them.

SEM Contrast and brightness slightly increased to correct the darkness issue.

Concerning the XRD, have you made any refinements to the diffractograms? If this is the case, the lattice parameters should be provided in the form of a table as additional information. You could also compare these values to the theoretical value of well crystallized CeO2. You could thus have possible information on the presence of Ce(III) or the incorporation of Nd and Eu in the fluorine structure.

The CeO2 films prepared for this work were unfortunately not crystalline enough to achieve reliable refinement from their diffractograms, and as such precise lattice parameters could not be determined.  This would be achievable for more crystalline (i..e higher preparation temp) or bulk samples, but isn't vialbe for the work here. Confirmation of the correct phase was validated through peak-matching using a comprehensive crystallography database which accompanies our diffractometer. SEM elemental mapping using EDAX validates the even distribution of the dopants throughout the films. No post-processing was applied to the XRD patterns presented in either the main text or SI.

- Concerning the leaching part, the increase in the dissolution rate of the irradiated compound (31%) is in fact perhaps due to the presence of Ce (III) but it is also probably due to the presence of cracks and therefore to an increase in the reactive surface of the sample.

Comment added to main text to address this. Thank you for raising this!

Generally speaking, when we look at Figures S11 and S12, the release of cations into solution takes place during the first day. This pulse (already seen in the literature) is generally due to the presence of a poorly crystallized phase (therefore very soluble), which is very likely for samples calcined at 200 °C. I think it would be better to add a few words on this effect in the text. Indeed the difference in behavior between doped Nd and Eu is very surprising, do they have the same crystallization state initially?

The Eu- and Nd-doped samples all show similar crystallinity (with respect to Scherrer crystallite size) to each other and the pure CeO2 film. It may be that the Eu and Nd nitrates are not fully calcined in the same manner as CAN, but determining this would require access to analytical techniques beyond the suitability of these samples for analysis – their semi-frangible and fragile nature limits the use of some techniques.

Reviewer 2 Report

Comments and Suggestions for Authors

The paper was prepared professionally. The authors have deep knowledge of the topic. The paper contains all the necessary elements of the peer-reviewed paper.

Minor comments:

The term “reactivity” may have at least two meanings, i.e. chemical reactivity and reactivity of the nuclear fuel. Please clarify the meaning in the text (e.g. use the term “chemical reactivity”).

The quality of Figures 1, 2, and 5 should be improved. In addition, the Figure 2 should be more descriptive.

Please add a paragraph about the general characterization of the spent nuclear fuel with possible sources of information about irradiated fuel samples and related research like burnup validation, e.g. https://doi.org/10.3390/en15093041

Author Response

We thank the reviewer for taking the time to review our work and for their helpful suggestions.

Minor comments:

The term “reactivity” may have at least two meanings, i.e. chemical reactivity and reactivity of the nuclear fuel. Please clarify the meaning in the text (e.g. use the term “chemical reactivity”).

Corrected as requested.

The quality of Figures 1, 2, and 5 should be improved. In addition, the Figure 2 should be more descriptive.

Figure 1 reworked to be clearer.

We have clarified the text accompanying Figure 2 to better reflect the contents of the figure.

Figure 5 has been replaced with two better figures with greater resolution and colour for increased clarity.

 Please add a paragraph about the general characterization of the spent nuclear fuel with possible sources of information about irradiated fuel samples and related research like burnup validation, e.g. https://doi.org/10.3390/en15093041

Short discussion covering this topic added with additional, relevant references, in addition to that suggested.

Reviewer 3 Report

Comments and Suggestions for Authors

The paper contains interesting data on ceria-based thin films of SIMFUEL and is recommended to publish after minor amendments aiming to clarify some of uncertainties with data provided.

The amendment needed concerns data on leaching within Chapter 2.7 and corresponding plots within Figure S12: Ce (a and b), and Eu (c and d), leaching over time in Ce-Eu sample 562 matrix. Virgin (a, c) and irradiated (b, d). Leaching of nuclear waste forms including SNF considered as waste to be disposed of follows national and internationally accepted test protocols such as ISO, ASTM, etc. – see e.g. the IAEA publication https://www.iaea.org/publications/7655/strategy-and-methodology-for-radioactive-waste-characterization or characterisation overviews – https://doi.org/10.1016/B978-0-08-102702-8.00014-5 and https://doi.org/10.1038/s41529-021-00210-4.

From the texts within Chapter 2.7 and associated supplementary material it remains unspecified the main parameter – the normalised leaching rate. Could author clarify to readers how to link the concentrations of elements provided with expected normalised leaching rates?

Author Response

We thank the reviewer for taking the time to review our work and for their helpful suggestions.

The amendment needed concerns data on leaching within Chapter 2.7 and corresponding plots within Figure S12: Ce (a and b), and Eu (c and d), leaching over time in Ce-Eu sample 562 matrix. Virgin (a, c) and irradiated (b, d). Leaching of nuclear waste forms including SNF considered as waste to be disposed of follows national and internationally accepted test protocols such as ISO, ASTM, etc. – see e.g. the IAEA publication https://www.iaea.org/publications/7655/strategy-and-methodology-for-radioactive-waste-characterization or characterisation overviews – https://doi.org/10.1016/B978-0-08-102702-8.00014-5 and https://doi.org/10.1038/s41529-021-00210-4.

From the texts within Chapter 2.7 and associated supplementary material it remains unspecified the main parameter – the normalised leaching rate. Could author clarify to readers how to link the concentrations of elements provided with expected normalised leaching rates?

We have added a clarifying paragraph citing the suggested references and discussing the applicability of these to the nature of the samples prepared and explored in this work.

Reviewer 4 Report

Comments and Suggestions for Authors

The work is sound and well presented.

I only have editorial comments:

-why use "M" and not "m" for the mass differences (page 5 line 215)?

- when using unit, leave a space/dot between numerator and denominator, e.g. "g cm-3" instead of "gcm-3"

- p13 line 535 a space is missing

Author Response

We thank the reviewer for taking the time to review our work and for their helpful suggestions.

I only have editorial comments:

-why use "M" and not "m" for the mass differences (page 5 line 215)?

Changed as requested.

- when using unit, leave a space/dot between numerator and denominator, e.g. "g cm-3" instead of "gcm-3"

Corrected throughout.

- p13 line 535 a space is missing

Amended as suggested